# ANY-STEP DYNAMICS MODEL IMPROVES FUTURE PREDICTIONS FOR ONLINE AND OFFLINE REINFORCEMENT LEARNING

**Haoxin Lin**[1,2], **Yu-Yan Xu**[2], **Yihao Sun**[1], **Zhilong Zhang**[1,2], **Yi-Chen Li**[1,2],
**Chengxing Jia**[1,2], **Junyin Ye**[1,2], **Jiaji Zhang**[1], **Yang Yu**[1,2,3*]

[1]National Key Laboratory for Novel Software Technology, Nanjing University, Nanjing, China
& School of Artificial Intelligence, Nanjing University, Nanjing, China
[2]Polixir Technologies, Nanjing, China
[3]Peng Cheng Laboratory, Shenzhen, 518055, China
`linhx@lamda.nju.edu.cn,yuyan.xu@polixir.ai,`
`{sunyh,zhangzl,liyc,jiacx,yejy,zhangjj,yuy}@lamda.nju.edu.cn`

## ABSTRACT

Model-based methods in reinforcement learning offer a promising approach to enhance data efficiency by facilitating policy exploration within a dynamics model. However, accurately predicting sequential steps in the dynamics model remains a challenge due to the bootstrapping prediction, which attributes the next state to the prediction of the current state. This leads to accumulated errors during model roll-out. In this paper, we propose the **A**ny-step **D**ynamics **M**odel (ADM) to mitigate the compounding error by reducing bootstrapping prediction to direct prediction. ADM allows for the use of variable-length plans as inputs for predicting future states without frequent bootstrapping. We design two algorithms, ADMPO-ON and ADMPO-OFF, which apply ADM in online and offline model-based frameworks, respectively. In the online setting, ADMPO-ON demonstrates improved sample efficiency compared to previous state-of-the-art methods. In the offline setting, ADMPO-OFF not only demonstrates superior performance compared to recent state-of-the-art offline approaches but also offers better quantification of model uncertainty using only a single ADM. The code is available at https://github.com/LAMDA-RL/ADMPO.

## 1 INTRODUCTION

Model-based Reinforcement Learning (MBRL) (Luo et al., 2024b) has demonstrated empirical success in both online (Feinberg et al., 2018; Buckman et al., 2018; Chua et al., 2018; Luo et al., 2019; Janner et al., 2019; Lin et al., 2023) and offline (Yu et al., 2020; Kidambi et al., 2020; Yu et al., 2021; Rigter et al., 2022; Sun et al., 2023; Luo et al., 2024a; Chen et al., 2024; Zhang et al., 2024) settings. The essence of MBRL lies in the dynamics model, where extensive explorations and evaluations of the agent can occur, thereby reducing the reliance on real-world samples. Embedded in the model-based framework, online policy optimization can leverage a large Update-To-Data (UTD) ratio (Chen et al., 2021) to improve sample efficiency, while offline policy optimization can be completed using the model-augmented data beyond the dataset.

Although some efforts aim to propose high-fidelity dynamics models, such as adversarial models (Chen et al., 2023; Bhardwaj et al., 2023), causal models (Zhu et al., 2022), and ensemble dynamics models (Chua et al., 2018; Janner et al., 2019; Yu et al., 2020) adopted by the majority of MBRL algorithms, it is challenging to generate high-quality imaginary samples via long-horizon model roll-out. In a dynamics model with the common form, the state-action pair at time step $t$, $(s_t, a_t)$, is used as input to predict the next state $s_{t+1}$. Thus, the bootstrapping prediction, which attributes the next state to the prediction of the current state, is inevitably employed to roll out states in the dynamics model. The deviation of generated states increases with the roll-out length since the error

---
[*]Corresponding Author

accumulates gradually as the state transitions in imagination. If the policy is updated using unreliable samples with large compounding errors, it will be misled by biased policy gradients.

The impact of compounding error (Xu et al., 2021) on policy optimization restricts the utilization of the model, thereby hindering further improvements in the sample efficiency of online MBRL and the performance of offline MBRL. One potential way to deal with the issue of compounding error is to reduce bootstrapping prediction to direct prediction, considering the direct state transition after executing a multi-step action sequence (Asadi et al., 2018; 2019; Che et al., 2018; Machado et al., 2023). Although state $s_{t+1}$ is only dependent on state-action $(s_t, a_t)$ under the assumption of Markov property (Sutton & Barto, 2018), the prediction of $s_{t+1}$ can also leverage earlier information. Tracing back a prior $k$-step plan, $i.e.$, $s_{t+1-k}$ followed by a sequence of $k$-step actions $(a_{t+1-k}, a_{t+2-k}, \cdots, a_t)$, is sufficient to constitute an attribution to predict $s_{t+1}$.

However, a fixed-horizon multi-step dynamics model cannot allow for rolling out a sequence starting from a single state since the policy cannot make multi-step action decisions at once. More importantly, it still relies on an ensemble to estimate model uncertainty in the offline setting (Yu et al., 2020; Sun et al., 2023), which further increases the computational burden. To enhance multi-step direct prediction, we propose that a dynamics model should be capable of directly predicting future states across arbitrary horizon lengths. This would incorporate the flexible roll-out capacity of previous single-step dynamics models, and the variability in horizon lengths could create conditions for quantification of the model uncertainty, making it possible to abandon the ensemble.

To handle the variable-length plans, we introduce a special **A**ny-step **D**ynamics **M**odel (ADM) that allows for the use of $s_{t+1-k}$ and $(a_{t+1-k}, a_{t+2-k}, \cdots, a_t)$ corresponding to any integer $k$ within a specified range as inputs for predicting $s_{t+1}$. When the agent faces changes occurring in the trajectory distribution, the state predictions from different backtracking lengths will exhibit noticeable divergence. This feature naturally enables ADM to estimate model uncertainty without the ensemble. Replacing the ensemble dynamics model with ADM, we devise a unique model roll-out method with random backtracking, which can be plugged into any existing MBRL algorithmic framework. In this paper, our main purpose is to demonstrate how the augmented data generated by ADM exhibits excellent effectiveness, both in improving future predictions and measuring the model uncertainty.

In general, our contributions are summarized as follows. (1) We present a generalized dynamics model called ADM to replace the dynamics model used in existing online and offline MBRL algorithms and demonstrate its superiority in reducing compounding errors. (2) We propose a new online MBRL algorithm called ADMPO-ON based on ADM and show that it can outperform recent state-of-the-art online model-based algorithms in terms of sample efficiency while retaining competitive performance on MuJoCo (Todorov et al., 2012) benchmarks. (3) We propose a new offline MBRL algorithm called ADMPO-OFF based on ADM and show that it can effectively quantify the model uncertainty, achieving superior performance compared to recent state-of-the-art offline algorithms on D4RL (Fu et al., 2020) and NeoRL (Qin et al., 2022) benchmarks.

## 2 PRELIMINARIES

### 2.1 MARKOV DECISION PROCESS AND REINFORCEMENT LEARNING

We consider a standard Markov Decision Process (MDP) specified by a tuple $\mathcal{M} = (\mathcal{S}, \mathcal{A}, T, \rho_0, \gamma)$, where $\mathcal{S}$ is the state space, $\mathcal{A}$ is the action space, $T(s_{t+1}, r_{t+1}|s_t, a_t)$ is the dynamics function that calculates the conditioned distribution of $s_{t+1} \in \mathcal{S}$ and $r_{t+1} \in \mathbb{R}$ given $(s_t, a_t)$, $\rho_0$ is the initial state distribution, and $\gamma$ is the discount factor. We use $\rho^\pi$ to denote the on-policy distribution over states induced by the dynamics function $T$ and the policy $\pi$. From a multi-step perspective, the attribution of state $s_{t+1}$ and reward $r_{t+1}$ can be traced back to the earlier $k$-step plan, $s_{t-k+1}$ along with the action sequence $a_{t-k+1:t} = (a_{t-k+1}, a_{t-k+2}, \cdots, a_t)$ in between. This relationship can be represented by the $k$-step dynamics model

$$T^k(s_{t+1}, r_{t+1}|s_{t-k+1}, a_{t-k+1:t}) = \sum_{(s_{t-k+2:t}, r_{t-k+2:t}) \in \mathcal{S}^{k-1} \times \mathbb{R}^{k-1}} \prod_{i=0}^{k-1} T(s_{t-i+1}, r_{t-i+1}|s_{t-i}, a_{t-i}).$$

(1)

We use $\Gamma_\pi^k(s_{t-k+1:t}, a_{t-k+1:t}|s_{t+1})$ to denote the distribution over $(s_{t-k+1:t}, a_{t-k+1:t})$ conditioned on $s_{t+1}$ induced by the dynamic function $T$ and the policy $\pi$.

The optimization goal of Reinforcement Learning (RL) is to find a policy $\pi$ that maximizes the expected discounted return $\mathbb{E}_{\rho^\pi}\left[\sum_{t=1}^\infty \gamma^{t-1} r_t\right]$. Such a policy can be derived from the estimation of the state-action value function $Q^\pi(s_t, a_t) = \mathbb{E}_{(s_{t+1}, r_{t+1}) \sim T(\cdot|s_t, a_t)}\left[r_{t+1} + \gamma V^\pi(s_{t+1})\right]$, where $V^\pi(s_{t+1}) = \mathbb{E}_{a_{t+1} \sim \pi(\cdot|s_{t+1})}\left[Q^\pi(s_{t+1}, a_{t+1})\right]$ is the state value function.

## 2.2 Model-based Reinforcement Learning

MBRL aims to find the optimal policy while transferring the agent's explorations and evaluations from the environment to the learned dynamics model. Given a dataset $\mathcal{D}_{\text{env}}$ collected via interaction in the real environment, the dynamics model $\hat{T}$ is typically trained to maximize the expected likelihood $\mathbb{E}_{(s_t, a_t, r_{t+1}, s_{t+1}) \sim \mathcal{D}_{\text{env}}}[\log \hat{T}(s_{t+1}, r_{t+1}|s_t, a_t)]$. The estimated dynamics model defines a surrogate MDP $\hat{\mathcal{M}} = (\mathcal{S}, \mathcal{A}, \hat{T}, \rho_0, \gamma)$. Then any RL algorithm can be used to recover the optimal policy with the augmented dataset $\mathcal{D}_{\text{env}} \cup \mathcal{D}_{\text{model}}$, where $\mathcal{D}_{\text{model}}$ is the synthetic data rolled out in $\hat{\mathcal{M}}$.

The above-mentioned paradigm is adopted by model-based policy optimization (MBPO) (Janner et al., 2019) and much of its follow-up work (Lin et al., 2023; Li et al., 2022; Pan et al., 2020; Clavera et al., 2020) in the online setting. These works don't need to consider the issue of model coverage, as the agent can explore online to fill in the regions where the dynamics model is uncertain. However, in the offline setting, the limited dataset causes $\hat{T}$ to cover only a part of the state-action space. Therefore, MOPO (Yu et al., 2020) and some of its subsequent offline MBRL algorithms (Kidambi et al., 2020; Sun et al., 2023) use the ensemble-based uncertainty as a penalty term in the reward function, allowing the agent to sample within safe regions of $\hat{T}$. This work can estimate the model uncertainty without ensemble models. A related approach is IVE (Filos et al., 2022), which also proposes an ensemble-free uncertainty estimator through the divergence of values obtained by applying the surrogate Bellman operator to the value function for different times.

## 3 Method

In this section, we propose a special **A**ny-step **D**ynamics **M**odel (ADM) to replace the mainstream ensemble dynamics models. ADM reduces bootstrapping prediction to direct prediction by backtracking variable-length plans. Applying ADM to existing MBRL frameworks for policy optimization, we introduce two algorithms, namely online ADMPO-ON and offline ADMPO-OFF.

### 3.1 Any-step Dynamics Model

Currently, the prevalent dynamics models typically operate on a single-step basis, with $s_t$ and $a_t$ as inputs to predict $s_{t+1}$ and $r_{t+1}$. In a broader context, dynamics models can also be multi-step (Asadi et al., 2018; 2019; Che et al., 2018), where inputs encompass $s_t$ along with a $k$-step sequence of actions $(a_t, a_{t+1}, \cdots, a_{t+k-1})$ to predict $s_{t+k}$ and $r_{t+k}$. To introduce flexibility in the backtracking length of the model, we further extend the definition of the multi-step dynamics model to allow $k$ to be any positive integer within a specified range, as delineated in Definition 3.1.

**Definition 3.1** (Any-step Dynamics Model). Given the maximum backtracking length $m$, an any-step dynamics model $\hat{T}(s_{t+k}, r_{t+k}|s_t, a_{t:t+k-1})$ is the distribution of $s_{t+k} \in \mathcal{S}$ and $r_{t+k} \in \mathbb{R}$ conditioned on the $k$-step plan $(s_t, a_{t:t+k-1}) = (s_t, a_t, a_{t+1}, \cdots, a_{t+k-1}) \in \mathcal{S} \times \mathcal{A}^k$, where $k$ can be any integer between $[1, m]$.

To handle inputs with variable step sizes, we utilize an RNN (Elman, 1990) with a GRU (Cho et al., 2014) cell to implement the any-step dynamics model, as depicted in the left part of Figure 1. Certainly, Transformer (Vaswani et al., 2017) is also a feasible choice, but we do not consider it because the model structure is beyond the scope of this study. Since the input state consists of only one step, while the action may be a sequence of multiple steps, we duplicate the state to match the length of the action sequence, and then sequentially feed it into the RNN. The input $(s_t, a_{t:t+k-1})$, after being represented by the RNN, yields the hidden $h_t^k$, which is then fed into an MLP to obtain the mean and standard deviation of $s_{t+k}$ and $r_{t+k}$, i.e., $(\boldsymbol{\mu}_{t+k}^s, \boldsymbol{\Sigma}_{t+k}^s)$ and $(\boldsymbol{\mu}_{t+k}^r, \boldsymbol{\Sigma}_{t+k}^r)$. Similar to previous model-based methods (Janner et al., 2019; Pan et al., 2020; Lin et al., 2023), we model the distributions of $s_{t+k}$ and $r_{t+k}$ as Gaussian distributions and predict them through sampling. We call the **A**ny-step **D**ynamics **M**odel as ADM and denote it as $\hat{T}_\theta(s_{t+k}, r_{t+k}|s_t, a_{t:t+k-1})$, where

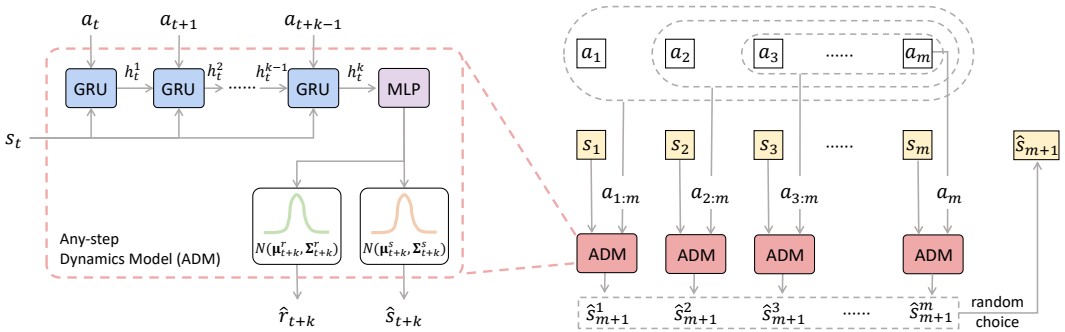

Figure 1: Illustration of any-step dynamics model (left) structured using RNN and its application for next-step prediction with random backtracking (right).

---

**Algorithm 1** Roll-out in ADM: **ADM-Roll**$(\hat{T}_\theta, \pi_\phi, H, m, (s_1, a_1, s_2, a_2, \cdots, s_{m-1}, a_{m-1}, s_m))$

---

**Input**: Learned ADM $\hat{T}_\theta$ with parameters $\theta$, policy $\pi_\phi$ with parameters $\phi$, roll-out length $H$, maximum backtracking length $m$, state-action sequence $(s_1, a_1, s_2, a_2, \cdots, s_{m-1}, a_{m-1}, s_m)$

1: **for** $\tau = 0$ to $H - 1$ **do**
2:    **if** $\tau = 0$ **then** Sample $a_{m+\tau} \sim \pi_\phi(\cdot|s_{m+\tau})$
3:    **else** Sample $a_{m+\tau} \sim \pi_\phi(\cdot|\hat{s}_{m+\tau})$
4:    Randomly sample an integer $k$ from $[1, m]$ uniformly
5:    **if** $\tau \leq k - 1$ **then** Roll out via $(\hat{s}_{m+\tau+1}, \hat{r}_{m+\tau+1}) \sim \hat{T}_\theta(\cdot|s_{m+\tau+1-k}, a_{m+\tau+1-k:m+\tau})$
6:    **else** Roll out via $(\hat{s}_{m+\tau+1}, \hat{r}_{m+\tau+1}) \sim \hat{T}_\theta(\cdot|\hat{s}_{m+\tau+1-k}, a_{m+\tau+1-k:m+\tau})$
7: **end for**
8: **return** $(s_m, a_m, \hat{r}_{m+1}, \hat{s}_{m+1}, \cdots, \hat{s}_{m+H-1}, a_{m+H-1}, \hat{r}_{m+H}, \hat{s}_{m+H})$

---

$\theta$ represents the neural parameters. With the real samples from the environment, $\hat{T}_\theta$ is trained to maximize the expected likelihood:

$$J_T(\theta) = \frac{1}{m} \sum_{k=1}^{m} \mathbb{E}_{(s_t, a_{t:t+k-1}, r_{t+k}, s_{t+k}) \sim \mathcal{D}_{\text{env}}} \left[ \log \hat{T}_\theta(s_{t+k}, r_{t+k} | s_t, a_{t:t+k-1}) \right]. \tag{2}$$

With $\hat{T}_\theta$, the frequent bootstrapping during model roll-out can be reduced. Specifically, given the maximum backtracking length $m$, a state-action sequence of length $m$, $(s_1, a_1, s_2, a_2, \cdots, s_m, a_m)$, is sampled from the data buffer to start the roll-out in $\hat{T}_\theta$. To obtain the prediction $\hat{s}_{m+1}$, an integer between $[1, m]$ is chosen uniformly at random as the backtracking length. If only one step is selected for backtracking, $(s_m, a_m)$ will be fed into $\hat{T}_\theta$ to obtain the prediction result; if $m-1$ steps are chosen, $(s_2, a_{2:m})$ will be fed into $\hat{T}_\theta$, and so forth. The right part of Figure 1 illustrates the aforementioned process based on random backtracking. Similarly, subsequent state predictions can also backtrack at most $m$ steps. After rolling out several steps, it inevitably backtracks to previously predicted states. For example, for the prediction $\hat{s}_{2m}$, it can only backtrack to one of $(s_m, \hat{s}_{m+1}, \hat{s}_{m+2}, \cdots, \hat{s}_{2m-1})$. The backtracked state, one part of the attribution for the next state prediction, is located several steps ahead, from the perspective of expectation. Thus, ADM reduces the actual bootstrapping count of a rolled-out trajectory. The complete $H$-step roll-out process in ADM is described in Algorithm 1.

Similar to existing MBRL algorithms, policy roll-out in ADM can generate a large number of fake samples for policy updates. We refer to the new dyna-style ADM-based policy optimization framework as ADMPO (**ADM**-based **P**olicy **O**ptimization). Any policy optimization algorithm can be plugged into this framework. In the subsequent subsections, we will introduce two new model-based algorithms, ADMPO-ON and ADMPO-OFF, for online and offline settings, respectively.

## 3.2 ADMPO-ON: ADM FOR POLICY OPTIMIZATION IN ONLINE SETTING

In the online setting, the agent interacts with the real environment while simultaneously optimizing the policy. Like MBPO (Janner et al., 2019), ADMPO-ON can be divided into two alternating stages, namely updating the dynamics model with continuously collected samples and utilizing samples generated through model roll-outs additionally for policy optimization. ADMPO-ON replaces the ensemble dynamics model in MBPO framework with ADM. It trains ADM with the optimization objective shown in Equation (2) and generates a large number of fake samples using the roll-out method depicted in Algorithm 1. The detailed pseudo-code is provided by Algorithm 2 in Appendix C.1.

During roll-outs, ADM randomly selects a backtracking length at each step and attributes the states to be predicted to variable-length plans. While backtracking $k$ steps, we view the sampling process $(\hat{s}_{t+1}, \hat{r}_{t+1}) \sim \hat{T}_\theta(\cdot|s_{t-k+1}, a_{t-k+1:t})$ as $(\hat{s}_{t+1}, \hat{r}_{t+1}) = \mu_\theta(s_{t-k+1}, a_{t-k+1:t}) + \eta_{t+1}$ with $\eta_{t+1} \sim \mathcal{N}(0, \Sigma_\theta(s_{t-k+1}, a_{t-k+1:t}))$, where $\mu_\theta$ is the deterministic dynamics function and $\Sigma_\theta$ is the standard deviation function used to construct the noise distribution with zero mean. In expectation, the target value of $Q(s_t, a_t)$ is estimated as

$$\mathbb{E}_{(s_{t-m+1:t-1}, a_{t-m+1:t-1}) \sim \Gamma_\pi^{m-1}(\cdot|s_t)} \left[ \frac{1}{m} \sum_{k=1}^{m} \mathbb{E}_{(\hat{s}_{t+1}, \hat{r}_{t+1}) \sim \hat{T}_\theta(\cdot|s_{t-k+1}, a_{t-k+1:t})} \left[ y(\hat{s}_{t+1}, \hat{r}_{t+1}) \right] \right],$$

(3)

where $y(\hat{s}_{t+1}, \hat{r}_{t+1}) = \hat{r}_{t+1} + \gamma \mathbb{E}_{a \sim \pi(\cdot|\hat{s}_{t+1})} [Q(\hat{s}_{t+1}, a)]$. Data generated via roll-outs in our ADM can be viewed as an implicit augmentation. The augmentation stems from two sources: (i) variation of the backtracking length while applying the learned ADM to predict the next state, and (ii) the noise introduced by the distribution $\mathcal{N}(0, \Sigma_\theta(s_{t-k+1}, a_{t-k+1:t}))$ at each backtracking length $k$. According to (Zheng et al., 2023), variations of state predictions can effectively implicitly regularize the local Lipschitz condition of the Q network around regions where the model prediction is uncertain, thereby regulating the value-aware model error (Farahmand et al., 2017).

## 3.3 ADMPO-OFF: ADM FOR POLICY OPTIMIZATION IN OFFLINE SETTING

In the offline setting, due to limitations of the behavior policy corresponding to the dataset, the learned ADM can only cover some regions of the state-action space. Beyond these safe regions lie the risky regions where the model is uncertain and unable to be fixed since online exploration is inaccessible to the agent. To prevent policy optimization collapse, exploitation of the learned model needs to be focused within the safe regions. Simultaneously, efforts should be made to explore beyond the boundaries of the safe regions to discover samples conducive to a better policy than the behavior policy. Achieving such a balance between conservatism and generalization often requires measuring model uncertainty. Based on ADM, we will introduce a new uncertainty quantification method.

In our ADM, states predicted using different backtracking lengths exhibit discrepancies. Intuitively, these discrepancies are closely related to the data distribution. When the agent is in safe regions, the discrepancies are small. As the agent gradually moves toward risky regions, the discrepancies tend to increase. The difference among probabilistic predictions $\hat{T}_\theta(\cdot|s_{t-k+1}, a_{t-k+1:t})$ obtained with different backtracking $k$ serve as a natural measure of model uncertainty, which can be quantified using variance (or standard deviation), as defined by Definition 3.2.

**Definition 3.2** (ADM-Uncertainty Quantifier). For any maximum backtracking length $m$ and the corresponding learned ADM $\hat{T}_\theta$, the uncertainty of $\hat{T}_\theta$ at $(s_t, a_t)$ is quantified as

$$\mathcal{U}^{\text{ADM}}(s_t, a_t) = \mathbb{E}_{\Gamma_\pi^{m-1}(\cdot|s_t)} \left[ \left\| \text{Var}_{k \sim \text{Uniform}(m), \hat{s}_{t+1} \sim \hat{T}_\theta(\cdot|s_{t-k+1}, a_{t-k+1:t})} [\hat{s}_{t+1}] \right\|_1 \right]$$

$$= \mathbb{E}_{\Gamma_\pi^{m-1}(\cdot|s_t)} \left[ \left\| \frac{1}{m} \sum_{k=1}^{m} \left( (\Sigma_\theta^k)^2 + (\mu_\theta^k)^2 \right) - (\bar{\mu})^2 \right\|_1 \right]$$

(4)

for any $s_t \in \mathcal{S}$ and $a_t \in \mathcal{A}$, where $\Sigma_\theta^k = \Sigma_\theta(s_{t-k+1}, a_{t-k+1:t})$, $\mu_\theta^k = \mu_\theta(s_{t-k+1}, a_{t-k+1:t})$ for convenience, and $\bar{\mu} = \frac{1}{m} \sum_{k=1}^{m} \mu_\theta^k$.

This uncertainty term corresponds to a combination of epistemic and aleatoric model uncertainty with a similar form to the ensemble standard deviation (Lu et al., 2022; Lakshminarayanan et al., 2017).

However, the source of diversity has shifted from ensemble to variable backtracking lengths. Since estimating the approximation error via epistemic or aleatoric uncertainty has been applied in many works (Yu et al., 2020; Bai et al., 2022; Sun et al., 2023; Lu et al., 2022), we assume that our ADM uncertainty (4) is an admissible error estimator (Yu et al., 2020), as described in Assumption 3.3.

**Assumption 3.3** (Admissible Error Estimator). Assume that there exists a positive $b \in \mathbb{R}^+$ such that the following inequality (5) holds for any maximum backtracking length $m$ and any $s_t \in \mathcal{S}, a_t \in \mathcal{A}$.

$$D_{\mathrm{TV}}(\bar{T}_{\theta,m}(\cdot|s_t, a_t), T(\cdot|s_t, a_t)) \leq b \cdot \mathcal{U}^{\mathrm{ADM}}(s_t, a_t), \tag{5}$$

where $\bar{T}_{\theta,m}$ is the overall conditioned distribution coming from

$$\bar{T}_{\theta,m}(\cdot|s_t, a_t) = \frac{1}{m} \sum_{k=1}^{m} \left[ \sum_{\substack{s_{t-k+1} \\ a_{t-k+1:t-1}}} \Gamma_{\pi}^{k-1}(s_{t-k+1}, a_{t-k+1:t-1}|s_t)\hat{T}_{\theta}(\cdot|s_{t-k+1}, a_{t-k+1:t})) \right]. \tag{6}$$

Under Assumption 3.3 and the $\xi$-uncertainty quantifier definition (see Appendix A for details) proposed by PEVI (Jin et al., 2021), we present the following theorem, demonstrating that $\mathcal{U}^{\mathrm{ADM}}$ can serve as a $\xi$-uncertainty quantifier to bound the Bellman error.

**Theorem 3.4.** $\beta \cdot \mathcal{U}^{\mathrm{ADM}}$ *is a valid $\xi$-uncertainty quantifier, with $\beta = b\frac{\gamma r_{\max}}{1-\gamma}$. Specifically,*

$$\left| \hat{\mathcal{T}}^{\pi} Q(s_t, a_t) - \mathcal{T}^{\pi} Q(s_t, a_t) \right| \leq \beta \cdot \mathcal{U}^{\mathrm{ADM}}(s_t, a_t), \tag{7}$$

*where $\hat{\mathcal{T}}^{\pi}$ is the proxy Bellman operator induced by ADM to estimate the true Bellman operator $\mathcal{T}^{\pi}$.*

*Proof.* See Appendix B. $\square$

According to the suboptimality theorem (see Appendix A for details) presented by PEVI (Jin et al., 2021), the policy $\hat{\pi}$ derived via pessimistic value iteration, which incorporates any $\xi$-uncertainty quantifier as a penalty term into the value iteration process (Sutton & Barto, 2018), has a bounded optimality gap to the optimal policy $\pi^*$. The optimality gap is dominated by the Bellman error and the uncertainty quantification. Intuitively, the Bellman error is usually small in safe regions where the dynamics model has been trained with rich data and tends to yield high consistency under different backtracking lengths, while large errors often appear in risky regions where data is scarce and the predictions via backtracking different lengths become inconsistent. The penalization prevents the policy from taking actions leading it to risky regions, thus the model cannot induce inaccurate value estimations on these actions. Thus, we can penalize the Bellman operator to obtain a pessimistic value estimation by

$$\hat{\mathcal{T}}^{\mathrm{ADM}}Q(s_t, a_t) := \hat{\mathcal{T}}^{\pi} Q(s_t, a_t) - \beta \cdot \mathcal{U}^{\mathrm{ADM}}(s_t, a_t). \tag{8}$$

We expect the penalty term $\beta \cdot \mathcal{U}^{\mathrm{ADM}}(s_t, a_t)$ to be as small as possible thereby constraining the optimality gap. While our Assumption 3.3 lacks theoretical guarantees and the tightness of the bound in Theorem 3.4 is unclear, we have provided sufficient evidence in Section 4.3.3 that our uncertainty quantification effectively estimates the model error.

Overall, ADMPO-OFF is the offline version of ADMPO-ON, which introduces a penalized Bellman operator (8) into the policy optimization process of ADMPO-ON, following the algorithmic framework of MOPO (Yu et al., 2020). The detailed pseudo-code is provided by Algorithm 3 in Appendix C.2.

## 4 EXPERIMENTS

In this section, we conduct several experiments to answer: (1) Does ADM roll out samples with less compounding error than the ensemble dynamics model? (2) How well does ADMPO-ON perform in the online setting? (3) How well does ADMPO-OFF perform in the offline setting? Does ADM quantify the model uncertainty better than the ensemble dynamics model?

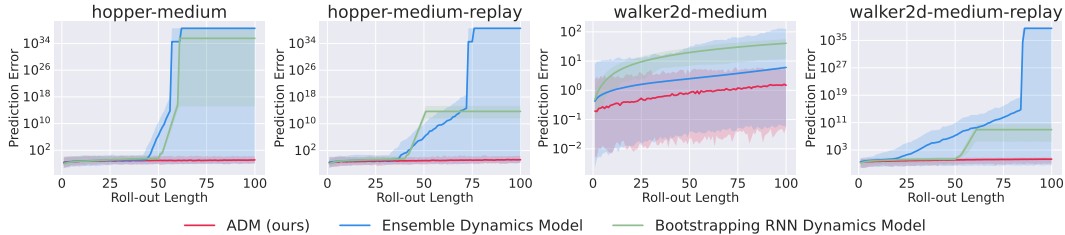

Figure 2: Comparison among ADM, ensemble dynamics model, and bootstrapping RNN dynamics model, in terms of the growth curve of the compounding error (in log scale) as roll-out length increases, after offline learning. The overflow value is regarded as the maximum value of `float32`.

## 4.1 DYNAMICS MODEL EVALUATION

An essential metric for evaluating dynamics model quality is the compounding error, which increases with the roll-out length. We selected four D4RL (Fu et al., 2020) datasets, hopper-medium-v2, hopper-medium-replay-v2, walker2d-medium-v2, and walker2d-medium-replay-v2, to compare the compounding error between ADM and the commonly used ensemble dynamics model. To eliminate the influence of the RNN structure, we also compare the bootstrapping RNN dynamic model, which shares the same structure as ADM but makes the prediction $\hat{s}_{t+1}$ using the historical state-action sequence $(\hat{s}_{t-k+1}, a_{t-k+1}, \cdots, \hat{s}_t, a_t)$[1] as input, where $k$ is uniformly sampled from $[1, m]$. The dataset is divided into a training set and a validation set, with the former used to train the dynamics model and the latter used to evaluate the compounding error. Figure 2 shows the growth curves of the compounding error as the roll-out length increases, with the y-axis in the log scale. The linear-scale version is shown in Appendix E.1. We observe that the curves of ADM remain close to zero, while the other two models exhibit exponential growth as the roll-out length exceeds a certain threshold. This phenomenon suggests ADM can improve predictions for future states due to its any-step backtracking mechanism during model roll-outs. We compare the performance of offline RL training using ADM and the bootstrapping RNN model in Appendix E.4.

## 4.2 EVALUATION IN ONLINE SETTING

We evaluate ADMPO-ON on four difficult MuJoCo continuous control tasks (Todorov et al., 2012), including Hopper, Walker2d, Ant, and Humanoid. All the tasks adopt version v3 and follow the default settings. Five model-based methods and one model-free method are selected as our baselines. These include SAC (Haarnoja et al., 2018), which is the state-of-the-art model-free RL algorithm; STEVE (Buckman et al., 2018), which incorporates an ensemble into the model-based value expansion; MBPO (Janner et al., 2019), which updates the policy with a mixture of real environmental samples and branched roll-out data; BMPO (Lai et al., 2020), which builds upon MBPO and replaces the dynamics model with a bidirectional one; DDPPO (Li et al., 2022), which adopts a two-model-based learning method to control the prediction error and the gradient error; and MACURA (Frauenknecht et al., 2024), which uses inherent model uncertainty to consider local accuracy to make roll-out.

Figure 3 shows the learning curves of ADMPO-ON and six baselines, along with SAC's asymptotic performance. ADMPO-ON achieves competitive performance after fewer environmental steps than most of the baselines. Taking the most difficult Humanoid as an example, ADMPO-ON and MACURA have achieved 100% of SAC convergence performance (about 6000) after 150k steps, while DDPPO needs about 200k steps, and the other four methods can't get close to the blue dashed line even at step 300k. ADMPO-ON matches the performance of the previous state-of-the-art algorithm, MACURA, and dominates other baselines in terms of learning efficiency on the Humanoid task. After training, ADMPO-ON can achieve a final performance close to the asymptotic performance of SAC on all these four MuJoCo tasks. These results demonstrate that ADMPO-ON has both high sample efficiency and competitive performance. Further study on why ADMPO-ON performs well in the online setting can be found in Appendix E.2.

---

[1]It would be $(s_{t-k+1}, a_{t-k+1}, \cdots, s_t, a_t)$ from the dataset if it were at the beginning of the roll-out.

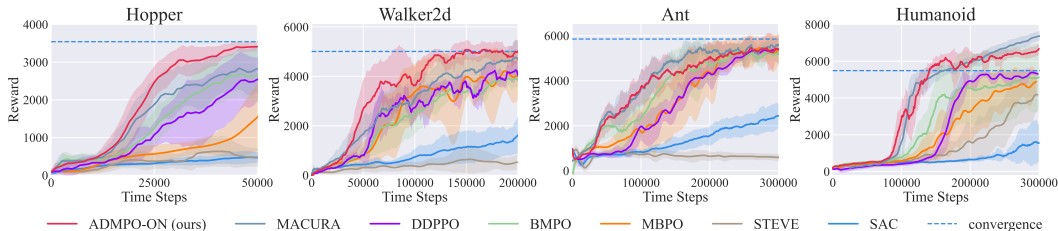

Figure 3: Online learning curves of ADMPO-ON (red) and other six baselines on four MuJoCo-v3 tasks. The blue dashed lines indicate the asymptotic performance of SAC for reference. The solid lines indicate the mean while the shaded areas indicate the standard error over five different seeds.

Table 1: Normalized scores after offline learning on D4RL MuJoCo tasks, averaged over five seeds.

| Task Name | BC | CQL | TD3+BC | EDAC | MOPO | COMBO | RAMBO | CBOP | MOBILE | ADMPO-OFF (ours) |
|---|---|---|---|---|---|---|---|---|---|---|
| hopper-random | 3.7 | 5.3 | 8.5 | 25.3 | 31.7 | 17.9 | 25.4 | 31.4 | 31.9 | **32.7±0.2** |
| halfcheetah-random | 2.2 | 31.3 | 11.0 | 28.4 | 38.5 | 38.8 | 39.5 | 32.8 | 39.3 | **45.4±2.8** |
| walker2d-random | 1.3 | 5.4 | 1.6 | 16.6 | 7.4 | 7.0 | 0.0 | 17.8 | 17.9 | **22.2±0.2** |
| hopper-medium | 54.1 | 61.9 | 59.3 | 101.6 | 62.8 | 97.2 | 87.0 | 102.6 | 106.6 | **107.4±0.6** |
| halfcheetah-medium | 43.2 | 46.9 | 48.3 | 65.9 | 73.0 | 54.2 | **77.9** | 74.3 | 74.6 | 72.2±0.6 |
| walker2d-medium | 70.9 | 79.5 | 83.7 | 92.5 | 84.1 | 81.9 | 84.9 | **95.5** | 87.7 | 93.2±1.1 |
| hopper-medium-replay | 16.6 | 86.3 | 60.9 | 101.0 | 103.5 | 89.5 | 99.5 | **104.3** | 103.9 | **104.4±0.4** |
| halfcheetah-medium-replay | 37.6 | 45.3 | 44.6 | 61.3 | **72.1** | 55.1 | 68.7 | 66.4 | 71.7 | 67.6±3.4 |
| walker2d-medium-replay | 20.3 | 76.8 | 81.8 | 87.1 | 85.6 | 56.0 | 89.2 | 92.7 | 89.9 | **95.6±2.1** |
| hopper-medium-expert | 53.9 | 96.9 | 98.0 | 110.7 | 81.6 | 111.1 | 88.2 | 111.6 | **112.6** | **112.7±0.3** |
| halfcheetah-medium-expert | 44.0 | 95.0 | 90.7 | 106.3 | 90.8 | 90.0 | 95.4 | 105.4 | **108.2** | 103.7±0.2 |
| walker2d-medium-expert | 90.1 | 109.1 | 110.1 | 114.7 | 112.9 | 103.3 | 56.7 | **117.2** | 115.2 | 114.9±0.3 |
| Average | 36.5 | 61.6 | 58.2 | 76.0 | 70.3 | 66.8 | 67.7 | 79.3 | 80.0 | **81.0** |

## 4.3 EVALUATION IN OFFLINE SETTING

### 4.3.1 D4RL BENCHMARK RESULTS

We compare ADMPO-OFF with four model-free methods: BC (behavioral cloning), which simply imitates the behavior policy of the dataset; CQL (Kumar et al., 2020), which equally penalized the Q values on out-of-the-distribution state-action pairs; TD3+BC (Fujimoto & Gu, 2021), which simply incorporates a BC term into the policy optimization objective of TD3 (Fujimoto et al., 2018); and EDAC (An et al., 2021), which quantifies the Q uncertainty via ensemble; as well as five model-based methods: MOPO (Yu et al., 2020), which adds the uncertainty of the model prediction as a penalization term to the reward function; COMBO (Yu et al., 2021), which introduces the penalty function of CQL into the model-based framework; RAMBO (Rigter et al., 2022), which adversarially trains the dynamics model and the policy; CBOP (Jeong et al., 2023), which adopts the variance of values under an ensemble of dynamics models to estimate the Q value conservatively under MVE (Feinberg et al., 2018) regime; and MOBILE (Sun et al., 2023), which proposes Model-Bellman inconsistency to estimate the Bellman error.

Table 1 reports the results on twelve D4RL (Fu et al., 2020) MuJoCo datasets (v2 version). The normalized score for each dataset is obtained via online evaluation after offline learning. The source of the reported performance is provided in Appendix D.5. We observe that ADMPO-OFF outperforms the other nine baselines in most tasks and achieves the highest average score. Notably, ADMPO-OFF has a significant performance advantage over MOPO. This directly demonstrates the effectiveness of ADM, since ADMPO-OFF only replaces the dynamics model in MOPO with ADM.

### 4.3.2 NEORL BENCHMARK RESULTS

NeoRL (Qin et al., 2022), is an offline RL benchmark that collects the data in a manner more conservative and closer to real-world data-collection scenarios. We focus on nine datasets collected using policies of three different qualities (low, medium, and high) in three environments Hopper-v3, HalfCheetah-v3, and Walker2d-v3, respectively. In our evaluation, each dataset contains 1000 trajectories.

Table 2: Normalized scores after offline learning on NeoRL tasks, averaged over five seeds.

| Task Name | BC | CQL | TD3+BC | EDAC | MOPO | MOBILE | ADMPO-OFF (ours) |
|---|---|---|---|---|---|---|---|
| neorl-hopper-low | 15.1 | 16.0 | 15.8 | 18.3 | 6.2 | 17.4 | **22.3±0.1** |
| neorl-halfcheetah-low | 29.1 | 38.2 | 30.0 | 31.3 | 40.1 | **54.7** | 52.8±1.2 |
| neorl-walker2d-low | 28.5 | 44.7 | 43.0 | 40.2 | 11.6 | 37.6 | **55.9±3.8** |
| neorl-hopper-medium | 51.3 | 64.5 | **70.3** | 44.9 | 1.0 | 51.1 | 51.5±5.0 |
| neorl-halfcheetah-medium | 49.0 | 54.6 | 52.3 | 54.9 | 62.3 | **77.8** | 69.3±1.7 |
| neorl-walker2d-medium | 48.7 | 57.3 | 58.5 | 57.6 | 39.9 | 62.2 | **70.1±2.4** |
| neorl-hopper-high | 43.1 | 76.6 | 75.3 | 52.5 | 11.5 | **87.8** | 87.6±4.9 |
| neorl-halfcheetah-high | 71.3 | 77.4 | 75.3 | 81.4 | 65.9 | 83.0 | **84.0±0.8** |
| neorl-walker2d-high | 72.6 | 75.3 | 69.6 | 75.5 | 18.0 | 74.9 | **82.2±1.9** |
| Average | 45.4 | 56.1 | 54.5 | 50.7 | 28.5 | 60.7 | **64.0** |

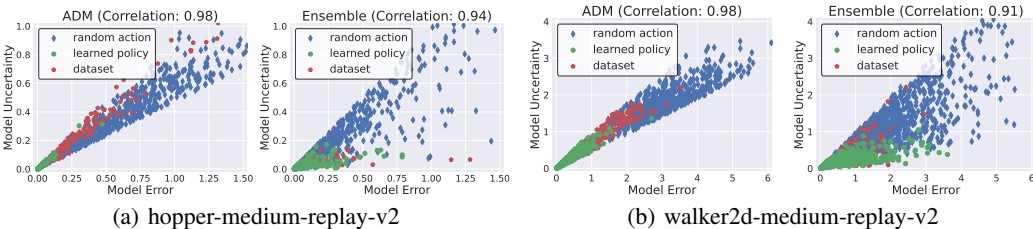

(a) hopper-medium-replay-v2        (b) walker2d-medium-replay-v2

Figure 4: Comparison between ADM and ensemble model in uncertainty quantification.

We compare our ADMPO-OFF with six baselines, including BC, CQL, TD3+BC, EDAC, MOPO, and MOBILE. Table 2 presents the normalized scores of these methods. Due to the narrow and limited coverage of the NeoRL data, all the baselines experience a decline in performance. In contrast, our ADMPO-OFF maintains a relatively high-level average score, still achieving superior performance in most tasks. This remarkable out-performance indicates the potential of our algorithm in more challenging real-world tasks.

### 4.3.3 UNCERTAINTY QUANTIFICATION

In our analysis, we sample a lot of state-action pairs in the learned ADM and the ensemble dynamics model respectively. These samples are obtained by model roll-out with three types of policy: random action selection, the learned policy after offline training, and the behavior policy of the dataset. Subsequently, we measure their model uncertainty and model error. The resulting scatter plots on two D4RL tasks, hopper-medium-replay-v2 and walker2d-medium-replay-v2, are illustrated in Figure 4. We observe that our ADM provides a better quantification for the model uncertainty. On the one hand, points sampled in ADM with greater model errors tend to exhibit greater quantified model uncertainty. The correlation coefficient of 0.98, observed across both tasks, surpasses that of the ensemble dynamics model. On the other hand, ADM can distinguish the samples from different policies better than the ensemble model. Samples generated from random actions deviate from the dataset distribution, whose uncertainty should be maximum in expectation. Conversely, when the learned policy is optimized within the safe regions covered by the dataset, model uncertainty is expected to be minimal. The experimental plots of ADM illustrate this phenomenon more clearly.

## 5 RELATED WORK

This work is related to online and offline dyna-style MBRL (Sutton, 1990).

### 5.1 ONLINE MODEL-BASED REINFORCEMENT LEARNING

In the online setting, MBRL algorithms aim to accelerate value estimation or policy optimization with model roll-out data. MVE (Feinberg et al., 2018) enhances Q-value target estimation by allowing short-term imagination to a fixed depth using the dynamics model. STEVE (Buckman et al., 2018) builds upon MVE by incorporating an ensemble into the value expansion to better estimate the Q

value. SLBO (Luo et al., 2019) directly utilizes TRPO (Schulman et al., 2015) to optimize the policy with synthetic data generated by rolling out to the end of trajectories in the dynamics model. MBPO (Janner et al., 2019) proposes a branched roll-out scheme to truncate unreliable samples, thereby reducing the influence of compounding error (Xu et al., 2021), and employs SAC (Haarnoja et al., 2018) to update the policy with a mixture of real-world data and model-generated data.

Recent work improves MBRL performance mainly from two perspectives. One focuses on learning a better dynamics model, such as bidirectional models (Lai et al., 2020), adversarial models (Chen et al., 2023; Bhardwaj et al., 2023), causal models (Zhu et al., 2022), multi-step models (Asadi et al., 2018; 2019; Che et al., 2018), and energy-based models (Chen et al., 2024). The other pursues a better utilization of the learned model, enhancing the reliability of model-generated samples (Pan et al., 2020) or applying model-based multi-step planning techniques (Chua et al., 2018; Clavera et al., 2020; Karkus et al., 2019; Okada et al., 2017; Srinivas et al., 2018; Lin et al., 2023).

## 5.2 Offline Model-based Reinforcement Learning

Although some model-free RL algorithms (Kumar et al., 2019; Fujimoto et al., 2019; Fujimoto & Gu, 2021; Kumar et al., 2020; An et al., 2021; Bai et al., 2022) have made significant contributions to offline RL research, MBRL algorithms appear to be more promising for the offline setting since they can utilize the dynamics model to extend the dataset and largely improve the data efficiency.

The core issue of offline MBRL lies in how to effectively leverage the model. MOPO (Yu et al., 2020) and MOReL (Kidambi et al., 2020) add the uncertainty of the model prediction as a penalization term to the original reward function to achieve a pessimistic value estimation. MOBILE (Sun et al., 2023) improves the uncertainty quantification by introducing Model-Bellman inconsistency into the offline model-based framework. COMBO (Yu et al., 2021) applies CQL (Kumar et al., 2020) to force the Q value to be small on model-generated out-of-distribution samples. RAMBO (Rigter et al., 2022) achieves conservatism by adversarial model learning for value minimization while keeping fitting the transition function. CBOP (Jeong et al., 2023) introduces adaptive weighting of short-horizon roll-out into MVE (Feinberg et al., 2018) technique and adopts the variance of values under an ensemble of dynamics models to estimate the Q value conservatively. MOREC (Luo et al., 2024a) designs a reward-consistent dynamics model using an adversarial discriminator to let the model-generated samples be more reliable.

## 6 Conclusion

In this work, we propose a new method for environment model learning and utilization, namely **A**ny-step **D**ynamics **M**odel (ADM). ADM is applicable in both online and offline MBRL frameworks, which yields two algorithms, ADMPO-ON and ADMPO-OFF, respectively. Several analyses and experiments show that ADM outperforms the ensemble dynamics model applied in previous MBRL approaches widely. Specifically, the compounding errors of multi-step roll-outs in ADM are much smaller than those in ensemble dynamics models. As a result, ADMPO-ON achieves significantly higher sampling efficiency in the online setting compared to MBPO. Additionally, the uncertainty term computed using ADM is more consistent with the model error, enabling ADMPO-OFF to achieve performance improvements in the offline setting. The remaining issues include that RNNs may consume more resources during the training process, and the effectiveness of ADM may be reduced in highly stochastic environments. These could serve as directions for future research. We believe ADM has powerful potential beyond the capabilities demonstrated in this paper. In the future, we will explore the scalability of ADM in non-Markovian visual RL scenarios, considering both online and offline settings.

## Acknowledgments

This work is supported by NSFC (62495093) and Jiangsu Science Foundation (BK20243039). The authors thank anonymous reviewers for their helpful discussions and suggestions for improving the article.

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

## A ADDITIONAL INTRODUCTION TO PESSIMISTIC VALUE ITERATION (PEVI)

Pessimistic Value Iteration (PEVI) (Jin et al., 2021) is a meta-algorithm for offline RL settings. It constructs an estimated Bellman operator $\hat{\mathcal{T}}^\pi$ based on the given dataset $\mathcal{D}_{\text{env}}$ to approximate the true Bellman operator $\mathcal{T}^\pi$ that satisfies

$$\mathcal{T}^\pi V_{h+1}(s_h, a_h) = \mathbb{E}_{(s_{h+1}, r_{h+1}) \sim T(\cdot | s_h, a_h)} \left[ r_{h+1} + V_{h+1}(s_{h+1}) \right], \tag{9}$$

where $h$ is the step index less than the horizon $\mathcal{H}$. Then the state-action value function is updated with

$$Q_h(s_h, a_h) \leftarrow \hat{\mathcal{T}}^\pi V_{h+1}(s_h, a_h) - \Lambda_h(s_h, a_h) \tag{10}$$

for each $(s_h, a_h)$, where $\Lambda_h$ is the penalty function that guarantees the conservatism of the learned policy. Especially, $\Lambda_h$ should be a $\xi$-uncertainty quantifier as follows.

**Definition A.1** ($\xi$-Uncertainty Quantifier (proposed by (Jin et al., 2021)))**.** The set of penalization $\{\Lambda_h\}_{h \in [\mathcal{H}]}$ forms a $\xi$-uncertainty quantifier if

$$\left| \hat{\mathcal{T}}^\pi V_{h+1}(s_h, a_h) - \mathcal{T}^\pi V_{h+1}(s_h, a_h) \right| \leq \Lambda_h(s_h, a_h) \tag{11}$$

holds with probability at least $1 - \xi$ for all $(s_h, a_h) \in \mathcal{S} \times \mathcal{A}$.

The following theorem characterizes the suboptimality of PEVI.

**Theorem A.2** (Suboptimality of PEVI (proposed by (Jin et al., 2021)))**.** *Suppose $\{\Lambda_h\}_{h=1}^{\mathcal{H}}$ in PEVI is a set of $\xi$-uncertainty quantifier. Then the derived policy $\hat{\pi}$ satisfies*

$$\left| V_1^{\pi^*}(s_1) - V_1^{\hat{\pi}}(s_1) \right| \leq 2 \sum_{h=1}^{\mathcal{H}} \mathbb{E}_{\rho^{\pi^*}} \left[ \Lambda_h(s_h, a_h) \right] \tag{12}$$

*with probability at least $1 - \xi$ for all starting $s_1 \in \mathcal{S}$. Here $\mathbb{E}_{\rho^{\pi^*}}$ is the expectation with respect to the trajectory induced by the optimal policy $\pi^*$ in the underlying MDP given the fixed function $\Lambda_h$.*

*Proof.* See PEVI (Jin et al., 2021) for detailed proof. $\square$

## B ADM UNCERTAINTY ANALYSIS

**Theorem B.1.** *$\beta \cdot \mathcal{U}^{\text{ADM}}$ is a valid $\xi$-uncertainty quantifier, with $\beta = b \frac{\gamma r_{\max}}{1 - \gamma}$. Specifically,*

$$\left| \hat{\mathcal{T}}^\pi Q(s_t, a_t) - \mathcal{T}^\pi Q(s_t, a_t) \right| \leq \beta \cdot \mathcal{U}^{\text{ADM}}(s_t, a_t), \tag{13}$$

*where $\hat{\mathcal{T}}^\pi$ is the proxy Bellman operator induced by ADM to estimate the true Bellman operator $\mathcal{T}^\pi$.*

*Proof.* First, we define $y(\hat{s}_{t+1}, \hat{r}_{t+1}) = \hat{r}_{t+1} + \gamma \mathbb{E}_{a \sim \pi(\cdot | \hat{s}_{t+1})} \left[ Q(\hat{s}_{t+1}, a) \right]$ and expand these two Bellman operator to

$$\hat{\mathcal{T}}^\pi Q(s_t, a_t)$$

$$= \mathbb{E}_{(s_{t-m+1:t-1}, a_{t-m+1:t-1}) \sim \Gamma_\pi^{m-1}(\cdot | s_t)} \left[ \frac{1}{m} \sum_{k=1}^{m} \mathbb{E}_{(\hat{s}_{t+1}, \hat{r}_{t+1}) \sim \hat{T}_\theta(\cdot | s_{t-k+1}, a_{t-k+1:t})} \left[ y(\hat{s}_{t+1}, \hat{r}_{t+1}) \right] \right]$$

$$= \sum_{\substack{s_{t-m+1} \\ a_{t-m+1:t-1}}} \Gamma_\pi^{m-1}(s_{t-m+1}, a_{t-m+1:t-1} | s_t) \left[ \frac{1}{m} \sum_{k=1}^{m} \sum_{\substack{\hat{s}_{t+1} \\ \hat{r}_{t+1}}} \hat{T}_\theta(\cdot | s_{t-k+1}, a_{t-k+1:t}) y(\hat{s}_{t+1}, \hat{r}_{t+1}) \right]$$

$$= \frac{1}{m} \sum_{k=1}^{m} \left[ \sum_{\substack{s_{t-k+1} \\ a_{t-k+1:t-1}}} \Gamma_\pi^{k-1}(s_{t-k+1}, a_{t-k+1:t-1} | s_t) \sum_{\substack{\hat{s}_{t+1} \\ \hat{r}_{t+1}}} \hat{T}_\theta(\cdot | s_{t-k+1}, a_{t-k+1:t})) y(\hat{s}_{t+1}, \hat{r}_{t+1}) \right]$$

$$= \sum_{\hat{s}_{t+1}, \hat{r}_{t+1}} \bar{T}_{\theta, m}(\hat{s}_{t+1}, \hat{r}_{t+1} | s_t, a_t) y(\hat{s}_{t+1}, \hat{r}_{t+1}),$$

$$\tag{14}$$

and

$$
\begin{aligned}
&\mathcal{T}^{\pi} Q(s_t, a_t) \\
&= \mathbb{E}_{\hat{s}_{t+1}, \hat{r}_{t+1} \sim T(\cdot | s_t, a_t)} \left[ y(\hat{s}_{t+1}, \hat{r}_{t+1}) \right] \\
&= \sum_{\hat{s}_{t+1}, \hat{r}_{t+1}} T(\hat{s}_{t+1}, \hat{r}_{t+1} | s_t, a_t) y(\hat{s}_{t+1}, \hat{r}_{t+1}).
\end{aligned} \tag{15}
$$

Then, we can obtain

$$
\begin{aligned}
&\left| \hat{\mathcal{T}}^{\pi} Q(s_t, a_t) - \mathcal{T}^{\pi} Q(s_t, a_t) \right| \\
&= \sum_{\hat{s}_{t+1}, \hat{r}_{t+1}} \left| \bar{T}_{\theta, m}(\hat{s}_{t+1}, \hat{r}_{t+1} | s_t, a_t) - T(\hat{s}_{t+1}, \hat{r}_{t+1} | s_t, a_t) \right| \cdot |y(\hat{s}_{t+1}, \hat{r}_{t+1})| \\
&= \gamma \sum_{\hat{s}_{t+1}, \hat{r}_{t+1}} \left| \bar{T}_{\theta, m}(\hat{s}_{t+1}, \hat{r}_{t+1} | s_t, a_t) - T(\hat{s}_{t+1}, \hat{r}_{t+1} | s_t, a_t) \right| \cdot \left| \mathbb{E}_{a \sim \pi(\cdot | \hat{s}_{t+1})} \left[ Q(\hat{s}_{t+1}, a) \right] \right| \\
&\leq \frac{\gamma r_{\max}}{1 - \gamma} \sum_{\hat{s}_{t+1}, \hat{r}_{t+1}} \left| \bar{T}_{\theta, m}(\hat{s}_{t+1}, \hat{r}_{t+1} | s_t, a_t) - T(\hat{s}_{t+1}, \hat{r}_{t+1} | s_t, a_t) \right| \\
&= \frac{\gamma r_{\max}}{1 - \gamma} D_{\mathrm{TV}}(\bar{T}_{\theta, m}(\cdot | s_t, a_t), T(\cdot | s_t, a_t)) \\
&\leq b \frac{\gamma r_{\max}}{1 - \gamma} \mathcal{U}^{\mathrm{ADM}}(s_t, a_t).
\end{aligned} \tag{16}
$$

Thus, let $\beta = b \frac{\gamma r_{\max}}{1-\gamma}$, we can say that $\beta \cdot \mathcal{U}^{\mathrm{ADM}}$ is a valid $\xi$-uncertainty quantifier, as defined by Definition A.1. $\qquad\square$

## C   Implementation Details

### C.1   ADMPO-ON

Our ADMPO-ON algorithm follows the framework of MBPO (Janner et al., 2019), as shown in Algorithm 2. The only difference between ADMPO-ON and MBPO lies in the way the dynamics model is trained and utilized, as indicated by the parts highlighted in blue in the pseudo-code.

---

**Algorithm 2** ADMPO-ON

---

**Input**: Initial ADM $\hat{T}_\theta$ and policy $\pi_\phi$, roll-out length $H$, maximum backtracking length $m$, real data buffer $\mathcal{D}_{\mathrm{env}}$, model data buffer $\mathcal{D}_{\mathrm{model}}$, warm-up size $U$, interaction epochs $N$, steps per epoch $E$

1: Explore for $U$ environmental steps and add data to $\mathcal{D}_{\mathrm{env}}$
2: **for** $N$ epochs **do**
3:     Train ADM $\hat{T}_\theta$ on $\mathcal{D}_{\mathrm{env}}$ by maximizing Equation (2)
4:     **for** $t = 1$ to $E$ **do**
5:         Sample action $a_t$ according to $\pi_\phi(\cdot | s_t)$
6:         Perform $a_t$ in the environment and add the real sample $(s_t, a_t, r_{t+1}, s_{t+1})$ to $\mathcal{D}_{\mathrm{env}}$
7:         **for** $M$ model roll-outs **do**
8:             Sample initial $m$-step state-action sequence $(s_{i:i+m-1}, a_{i:i+m-2})$ from $\mathcal{D}_{\mathrm{env}}$
9:             Roll out $H$ steps in $\hat{T}_\theta$ via **ADM-Roll**$(\hat{T}_\theta, \pi_\phi, H, m, (s_{i:i+m-1}, a_{i:i+m-2}))$ and add the model roll-out data to $\mathcal{D}_{\mathrm{model}}$
10:         **end for**
11:         **for** $G$ policy updates **do**
12:             Update current policy $\pi_\phi$ using samples from $\mathcal{D}_{\mathrm{env}} \cup \mathcal{D}_{\mathrm{model}}$
13:         **end for**
14:     **end for**
15: **end for**

---

## C.2 ADMPO-OFF

Our ADMPO-OFF algorithm follows the framework of MOPO (Yu et al., 2020), as shown in Algorithm 3. The only difference between ADMPO-OFF and MOPO lies in the way the dynamics model is trained and utilized, as indicated by the parts highlighted in blue in the pseudo-code.

---

**Algorithm 3** ADMPO-OFF

---

**Input**: Pre-collected dataset $\mathcal{D}_{\text{env}}$, initial ADM $\hat{T}_\theta$ and policy $\pi_\phi$, roll-out length $H$, maximum backtracking length $m$, model data buffer $\mathcal{D}_{\text{model}}$, iterations $N$, penalty coefficient $\beta$

1: Train ADM $\hat{T}_\theta$ on $\mathcal{D}_{\text{env}}$ by maximizing Equation (2)
2: **for** $N$ iterations **do**
3:    **for** $M$ model roll-outs **do**
4:       Sample initial $m$-step state-action sequence $(s_{i:i+m-1}, a_{i:i+m-2})$ from $\mathcal{D}_{\text{env}}$
5:       Roll out $H$ steps in $\hat{T}_\theta$ via **ADM-Roll**$(\hat{T}_\theta, \pi_\phi, H, m, (s_{i:i+m-1}, a_{i:i+m-2}))$
6:       Penalize the reward via $\tilde{r} = r - \beta \mathcal{U}^{\text{ADM}}(s, a)$ for each rolled-out step
7:       Add the penalized model roll-out data to $\mathcal{D}_{\text{model}}$
8:    **end for**
9:    **for** $G$ policy updates **do**
10:       Update current policy $\pi_\phi$ using samples from $\mathcal{D}_{\text{env}} \cup \mathcal{D}_{\text{model}}$
11:    **end for**
12: **end for**

---

## C.3 POLICY OPTIMIZATION

The policy optimization method used in our ADMPO-ON and ADMPO-OFF is SAC (Haarnoja et al., 2018), following MBPO (Janner et al., 2019) and MOPO (Yu et al., 2020). The hyper-parameters about SAC follow its standard implementation, as listed in Table 3.

Table 3: Hyper-parameters of Policy Optimization in ADMPO-ON and ADMPO-OFF.

| Hyper-parameter | Value | Description |
|---|---|---|
| $N_Q$ | 2 | the number of critics. |
| actor network | FC(256,256) | fully connected (FC) layers with ReLU activations. |
| critic network | FC(256,256) | fully connected (FC) layers with ReLU activations. |
| $\tau$ | $5 \times 10^{-3}$ | target network smoothing coefficient. |
| $\gamma$ | 0.99 | discount factor. |
| $lr_{\text{actor}}$ | $1 \times 10^{-4}$ | learning rate of actor. |
| $lr_{\text{critic}}$ | $3 \times 10^{-4}$ | learning rate of critic. |
| optimizer | Adam | optimizers of the actor and critics. |
| batch size | 256 | batch size for each update. |

# D EXPERIMENTAL DETAILS

## D.1 RESOURCE REQUIREMENTS

All experiments can be completed with just one NVIDIA GeForce RTX 2080 Ti or any other type of GPU with larger graphic memory. There are no additional resource requirements. The time of execution for each task is about 24 hours.

## D.2 ADMPO-ON SETTINGS

The experimental settings of our ADMPO-ON in Section 4.2 are listed in Table 4.

Table 4: Hyper-parameter settings of ADMPO-ON results presented in Figure 3. $x \to y$ over $a \to b$ denotes a thresholded linear increasing schedule, *i.e.* the length of model roll-outs at step $t$ is calculated by $f(t) = \min\left(\max\left(x + \frac{t-a}{b-a} \cdot (y-x), x\right), y\right)$.

| environment | Hopper | Walker2d | Ant | Humanoid |
|---|---|---|---|---|
| steps | 50k | 200k | 300k | |
| Update-To-Date ratio | 20 | | | |
| maximum backtracking length $m$ | 5 | 2 | | |
| model roll-out schedule | 1→15 over 0→50k | 1→10 over 0→100k | 1→5 over 10k→100k | 1→10 over 10k→100k |
| target entropy | -1 | -3 | -4 | -8 |

### D.3 ADMPO-OFF SETTINGS

The experimental settings of our ADMPO-OFF in Section 4.3 are listed in Table 5.

Table 5: Hyper-parameter settings of ADMPO-OFF results presented in Section 4.3.

| Domain Name | Task Name | $m$ | $H$ | $\beta$ |
|---|---|---|---|---|
| D4RL MuJoCo | hopper-random | 5 | 50 | 5 |
| | halfcheetah-random | 2 | 10 | 2.5 |
| | walker2d-random | 2 | 50 | 2.5 |
| | hopper-medium | 5 | 10 | 1 |
| | halfcheetah-medium | 2 | 5 | 2.5 |
| | walker2d-medium | 5 | 10 | 5 |
| | hopper-medium-replay | 5 | 5 | 0.1 |
| | halfcheetah-medium-replay | 2 | 5 | 2.5 |
| | walker2d-medium-replay | 5 | 5 | 0.1 |
| | hopper-medium-expert | 2 | 20 | 20 |
| | halfcheetah-medium-expert | 2 | 50 | 10 |
| | walker2d-medium-expert | 3 | 2 | 6 |
| NeoRL MuJoCo | neorl-hopper-low | 5 | 20 | 5 |
| | neorl-halfcheetah-low | 2 | 20 | 10 |
| | neorl-walker2d-low | 5 | 10 | 2.5 |
| | neorl-hopper-medium | 5 | 20 | 50 |
| | neorl-halfcheetah-medium | 2 | 5 | 20 |
| | neorl-Walker2d-medium | 5 | 10 | 5 |
| | neorl-hopper-high | 5 | 20 | 50 |
| | neorl-halfcheetah-high | 2 | 10 | 50 |
| | neorl-walker2d-high | 5 | 10 | 2.5 |

### D.4 HYPER-PARAMETER TUNING OF ADMPO-OFF

There are three important hyper-parameters in ADMPO-OFF: the maximum backtracking length $m$, the roll-out length $H$, and the penalty coefficient $\beta$. Below, we will introduce how to tune these three hyper-parameters.

- $m$: The first hyper-parameter to tune is $m$, as it can be adjusted based on the validation error after the training of the dynamics model. Experiments show that a moderately large $m$ can effectively ensure algorithm performance, and increasing $m$ does not lead to significant performance degradation. Considering that an excessively large $m$ would consume excessive computational resources, an initial value for m between 5 and 10 is recommended. Then, observe whether the validation error of the dynamics model after training meets the task requirements. If not, slightly decrease $m$, but ensure it does not go below 2.

- $H$: The second hyper-parameter to tune is $H$. After training the dynamics model, $H$ directly affects the compounding error generated during roll-outs using the model. A larger $H$ value can produce more diverse data, which benefits policy learning. It is recommended to set the initial value of $H$ to 5 and then gradually increase it until the compounding error after $H$-step roll-outs approaches the acceptable threshold.

- $\beta$: Tuning $\beta$ relies on the results of Q-value estimation. According to the findings of the previous work (Lu et al., 2022), in general, larger $H$ values require larger $\beta$ values for support. An appropriate initial $\beta$ value can be selected based on the size of $H$, typically ranging from one-tenth to one-half of $H$. Then, adjust $\beta$ based on the estimation bias of the Q-value. If the Q-value is overestimated, increase $\beta$; otherwise, decrease $\beta$. Finally, identify the critical $\beta$ value at which the Q-value is no longer overestimated.

### D.5 SOURCE OF BASELINES' RESULTS

For the evaluation on D4RL (Fu et al., 2020) benchmarks, the results of the compared baselines come from two sources:

- Retraining on D4RL datasets of v2 version with OfflineRL-Kit (Sun, 2023), for the algorithms whose original papers only report the performance on the v0 version, such as CQL (Kumar et al., 2020), MOPO (Yu et al., 2020).
- Including the scores in their papers, for the algorithms whose original papers report the performance on the v2 version, such as TD3+BC (Fujimoto & Gu, 2021), EDAC (An et al., 2021), RAMBO (Rigter et al., 2022), CBOP (Jeong et al., 2023), and MOBILE (Sun et al., 2023), or who does not provide source codes, such as COMBO (Yu et al., 2021).

For the evaluation on NeoRL (Qin et al., 2022) benchmarks, we report the scores of BC, CQL, and MOPO from the original paper of NeoRL and retrain TD3+BC and EDAC with OfflineRL-Kit (Sun, 2023).

## E ADDITIONAL EXPERIMENTS

### E.1 SUPPLEMENTARY RESULTS OF DYNAMICS MODEL EVALUATION

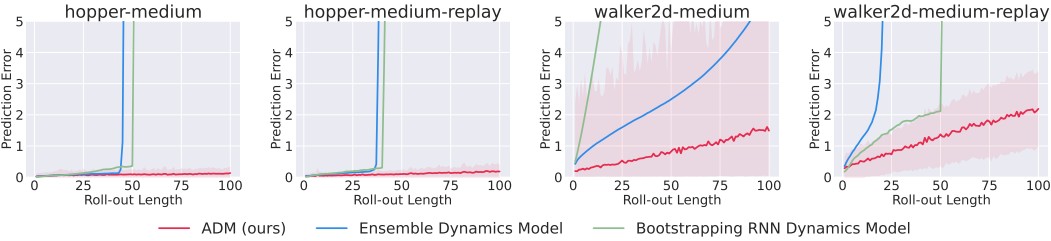

Figure 5: Comparison among ADM, ensemble dynamics model, and bootstrapping RNN dynamics model, in terms of the growth curve of the compounding error as roll-out length increases, after offline learning.

In Section 4.1, we present the compounding error curves of ADM, the bootstrapping RNN dynamics model, and the ensemble dynamics model as the roll-out length increases. The y-axis uses a log scale, which highlights ADM's advantage in future predictions distinctly. However, the log scale also makes it difficult to depict the precise values of ADM's compounding error. Here, we also provide the linear-scale version of Figure 2, as shown in Figure 5. In the hopper task, ADM's prediction error remains close to zero even after rolling out to 100 steps. In the more complex walker2d task, ADM's prediction error does not exceed 1 when the roll-out length is less than 50 steps.

Additionally, we set $m$ to 1, 5, 10, 15, and 20 respectively to observe the compounding error curves of ADM after offline training, as shown in Figure 6. Overall, as $m$ increases, the growth rate of

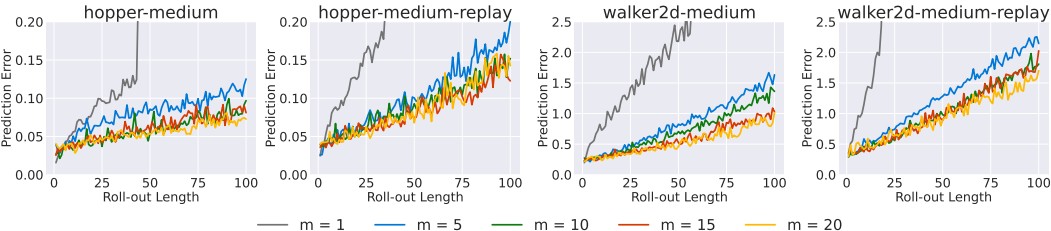

Figure 6: Comparison among ADM with $m$ set to 1, 5, 10, 15, 20, in terms of the growth curve of the compounding error as roll-out length increases, after offline learning.

the compounding error tends to slow down. This is because a larger $m$ implies fewer expected bootstrapping iterations and thus fewer error accumulation steps. Specifically, when $m = 1$, the growth rate of the compounding error is particularly rapid. This also indirectly demonstrates the effectiveness of ADM with an appropriate $m$.

### E.2    STUDY ON WHY ADMPO-ON PERFORMS WELL IN ONLINE SETTING

Value-aware model error (Farahmand et al., 2017) is a dependable metric for measuring the learning quality of the dynamics model and the suboptimality of the MBRL algorithm. We conduct a study to verify how well ADMPO-ON regulates the value-aware model error. Without loss of rigor, we only choose MBPO for comparison since most other model-based methods follow the same way of learning and utilizing the ensemble dynamics model. Figure 7 shows the results of the most difficult Humanoid task. The learned ADM in ADMPO-ON and the ensemble dynamics model in MBPO achieve similar mean squared errors, indicating their similar fitting abilities. However, ADMPO-ON provides greater model roll-out standard deviation over diverse state prediction, forcing the agent to explore more uncertain areas. Therefore, since the variation of state prediction helps smoothen the Q target, the Q network in ADMPO-ON has a significantly smaller Lipschitz constant, and then the value-aware model error, which measures the suboptimality of MBRL becomes smaller. This phenomenon explains why ADMPO-ON performs significantly better than MBPO in Figure 3. For details of the metrics used in this experiment, refer to (Zheng et al., 2023).

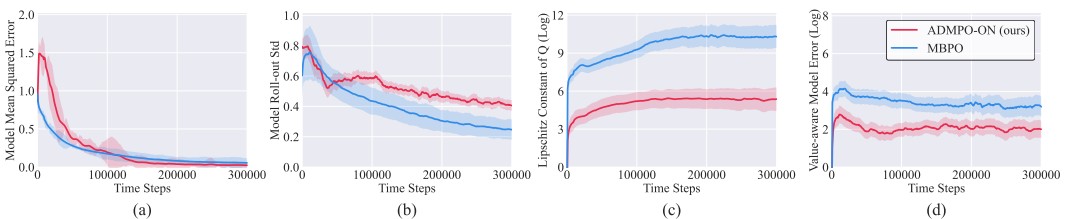

Figure 7: Comparison between ADMPO-ON and MBPO on Humanoid, in terms of (a) model mean squared error, (b) model roll-out standard deviation over diverse predictions, (c) estimated Lipschitz constant (Zheng et al., 2023) of Q, and (d) value-aware model error (Farahmand et al., 2017). Results are averaged over five seeds.

### E.3    D4RL ANTMAZE RESULTS

We compare ADMPO-OFF with several offline MBRL algorithms on the D4RL (Fu et al., 2020) AntMaze tasks, which test the capability of algorithms to find an effective goal-oriented policy. Table 6 reports the corresponding results. ADMPO-OFF still outperforms other offline MBRL baselines in most AntMaze tasks and achieves the highest average score. Notably, the policy learned by MOPO lacks any ability to reach the goal, scoring zero on all AntMaze tasks. However, by simply replacing the dynamics model with ADM, ADMPO-OFF can learn a policy with some capability to reach the goal, demonstrating potential in addressing such challenging tasks.

Table 6: Normalized scores after offline learning on D4RL AntMaze tasks, averaged over five seeds.

| Task Name | MOPO | COMBO | RAMBO | MOBILE | ADMPO-OFF (ours) |
|---|---|---|---|---|---|
| antmaze-umaze | 0.0 | 80.3 | 25.0 | 77.0 | **88.4±1.2** |
| antmaze-umaze-diverse | 0.0 | 57.3 | 0.0 | 20.4 | **81.7±8.6** |
| antmaze-medium-play | 0.0 | 0.0 | 16.4 | **64.6** | 23.9±6.3 |
| antmaze-medium-diverse | 0.0 | 0.0 | 23.2 | 1.6 | **24.1±5.7** |
| antmaze-large-play | 0.0 | 0.0 | 0.0 | 2.6 | **8.3±4.1** |
| antmaze-large-diverse | 0.0 | 0.0 | 2.4 | **7.2** | 0.0±0.0 |
| Average | 0.0 | 22.9 | 16.8 | 28.9 | **37.7** |

### E.4 STUDY ON DYNAMICS MODEL DESIGN

To predict the next state, ADM uniformly chooses a $k$ from $[1, m]$, then backtracks the state from $k$ steps earlier along with the following $k$-step sequence of actions as input. The prediction error of the next state is accumulated based on the deviation of the state from $k$ steps earlier. The only one-step input state is duplicated to match the length of the action sequence. To verify the effectiveness of ADM's design for policy optimization, we introduce two ablation versions of ADMPO-OFF:

- **BootRNN** replaces ADM with a bootstrapping RNN dynamics model, which uses the preceding $k$-step sequence of state-action pairs as input when predicting the next state, where $k$ is sampled uniformly from $[1, m]$. The prediction error of the next state is accumulated based on the deviation of the current state. All other settings remain consistent with ADMPO-OFF.

- **ADMPO-OFF-zero** takes a state and the following any-step actions as input to the dynamics model, the same as ADMPO-OFF. However, the state is only fed into the RNN during its first cell access, and subsequently, zero tensors are concatenated with the actions to match the dimensions. All other settings remain consistent with ADMPO-OFF.

We compare the performance of these dynamics model designs on nine D4RL Fu et al. (2020) MuJoCo tasks. Table 7 reports the corresponding results.

Table 7: Normalized scores corresponding to different dynamics model designs.

| Task Name | BootRNN | ADMPO-OFF-zero | ADMPO-OFF |
|---|---|---|---|
| hopper-medium | 89.8±1.8 | 106.0±0.6 | **107.4±0.6** |
| hopper-medium-replay | 87.2±2.7 | 101.2±0.1 | **104.4±0.4** |
| hopper-medium-expert | 108.6±1.0 | **112.9±1.5** | 112.7±0.3 |
| walker2d-medium | 78.9±1.4 | **94.1±0.2** | 93.2±1.1 |
| walker2d-medium-replay | 74.6±0.9 | 92.6±2.4 | **95.6±2.1** |
| walker2d-medium-expert | 95.5±2.3 | 113.2±0.2 | **114.9±0.3** |
| halfcheetah-medium | 52.8±1.2 | **73.7±0.1** | 72.2±0.6 |
| halfcheetah-medium-replay | 58.2±1.6 | 66.3±0.6 | **67.6±3.4** |
| halfcheetah-medium-expert | 94.0±1.2 | 101.1±0.3 | **103.7±0.2** |

BootRNN experiences significant policy performance degradation compared to ADMPO-OFF, since the bootstrapping RNN dynamics model accumulates the prediction error at each step based on the deviation from the last step, leading to a larger compounding error. This has already been validated in the dynamics model evaluation experiment in Section 4.1. ADMPO-OFF-zero does not exhibit significant performance differences compared to ADMPO-OFF, as their inputs are identical, differing only in the way they are forwarded. This difference has minimal impact on the overall performance.

### E.5 STUDY ON UNCERTAINTY QUANTIFIER

ADMPO-OFF uses the standard deviation of predictions as a measure of model uncertainty, which is actually a combination of epistemic and aleatoric model uncertainty. Other common uncertainty choices include maximum aleatoric uncertainty and maximum pairwise difference. Denoting the mean and standard deviation of predictions when backtracking $k$ steps as $\mu_\theta^k$ and $\Sigma_\theta^k$, respectively, the descriptions for these uncertainties are as follows.

- **Max aleatoric (Yu et al., 2020):** $\max_{k=1,\cdots,m} \|\Sigma_\theta^k\|_F$, which corresponds to the maximum aleatoric error.
- **Max Pairwise Difference (Kidambi et al., 2020):** $\max_{i,j} \|\mu_\theta^i - \mu_\theta^j\|_2$, which corresponds to the pairwise maximum difference of the predictions from different $k$.
- **Prediction Standard Deviation (Lu et al., 2022):** $\frac{1}{m} \sum_{k=1}^m \left( (\Sigma_\theta^k)^2 + (\mu_\theta^k)^2 \right) - \left( \frac{1}{m} \sum_{k=1}^m \mu_\theta^k \right)^2$, which corresponds to a combination of epistemic and aleatoric model uncertainty.

We compare the performance of these uncertainty choices on nine D4RL (Fu et al., 2020) MuJoCo tasks. Table 8 reports the corresponding results.

Table 8: Normalized scores corresponding to different uncertainty quantifier choices.

| Task Name | Max Aleatoric | Max Pairwise Difference | Prediction Std (ADMPO-OFF) |
|---|---|---|---|
| hopper-medium | 55.6±0.9 | 105.6±0.2 | **107.4±0.6** |
| hopper-medium-replay | 103.3±0.1 | 101.8±1.3 | **104.4±0.4** |
| hopper-medium-expert | **113.1±0.2** | 111.7±1.2 | 112.7±0.3 |
| walker2d-medium | 87.7±0.5 | 79.7±1.8 | **93.2±1.1** |
| walker2d-medium-replay | 93.4±0.3 | **96.3±0.9** | 95.6±2.1 |
| walker2d-medium-expert | 111.6±0.4 | 110.4±0.7 | **114.9±0.3** |
| halfcheetah-medium | 71.5±0.3 | 70.4±3.4 | **72.2±0.6** |
| halfcheetah-medium-replay | **69.7±1.3** | 59.4±2.8 | 67.6±3.4 |
| halfcheetah-medium-expert | 102.8±2.3 | 96.9±2.9 | **103.7±0.2** |

The prediction standard deviation, which is similar to the ensemble standard deviation in the ensemble dynamics model, performs the best. Similarly, it is found by (Lu et al., 2022) that ensemble standard deviation is the best while using the ensemble dynamics model. The key to ADM is that it provides a method for diversifying state predictions that is different from ensemble methods, and any form of uncertainty is feasible.

### E.6 STUDY ON PREDICTION CHOICE

While rolling out a sequence using ADM, we choose the backtracking length $k$ uniformly from $\{1, 2, \cdots, m\}$ then predict the next state by feeding the state from $k$ steps ago and the following $k$-step sequence of actions into the ADM. The random sampling of the backtracking length can be viewed as an implicit augmentation. The variations of state predictions can effectively implicitly regularize the local Lipschitz condition of the Q network around regions where the model prediction is uncertain, thereby regulating the value-aware model error (Farahmand et al., 2017), according to (Zheng et al., 2023). Other intuitive choices to predict the next state include:

- **Priority Sampling**: Determining the sampling priority based on the fitting losses of different backtracking lengths. For instance, the sampling probability corresponding to $k$ is given by $p_k = \frac{e^{-\epsilon_k}}{\sum_{i=1}^m e^{-\epsilon_i}}$, where $\epsilon_k$ is the fitting loss of the $k$-step prediction.
- **Max Backtracking**: Always choosing the maximum backtracking length $m$ to predict the next state.
- **Average Prediction**: Averaging the predictions corresponding to different backtracking lengths from $\{1, 2, \cdots, m\}$.

We compare the performance of these prediction choices on nine D4RL (Fu et al., 2020) MuJoCo tasks. Table 9 reports the corresponding results.

Table 9: Normalized scores corresponding to different prediction choices.

| Task Name | Priority Sampling | Max Backtracking | Average Prediction | Uniform (ADMPO-OFF) |
|---|---|---|---|---|
| hopper-medium | 106.4±0.8 | 98.8±0.9 | 56.1±2.4 | **107.4±0.6** |
| hopper-medium-replay | **105.3±0.8** | 95.6±0.8 | 51.6±1.9 | 104.4±0.4 |
| hopper-medium-expert | **113.6±3.7** | 110.3±1.6 | 52.6±1.4 | 112.7±0.3 |
| walker2d-medium | 89.0±1.5 | 84.5±1.1 | 85.9±1.8 | **93.2±1.1** |
| walker2d-medium-replay | **96.2±1.0** | 91.1±0.6 | 87.0±1.6 | 95.6±2.1 |
| walker2d-medium-expert | **115.5±0.7** | 107.7±1.1 | 88.6±1.0 | 114.9±0.3 |
| halfcheetah-medium | 71.4±3.2 | 56.9±0.2 | 58.9±1.3 | **72.2±0.6** |
| halfcheetah-medium-replay | 61.7±2.3 | 49.8±0.9 | 46.8±1.3 | **67.6±3.4** |
| halfcheetah-medium-expert | 103.2±1.4 | 67.0±1.4 | 63.2±0.9 | **103.7±0.2** |

We find that there is no significant performance difference between the priority sampling and the uniform sampling, since both of them introduce diversity in state prediction. Both maximum backtracking and average prediction experience significant performance degradation.

### E.7 STUDY ON $m$

We set $m$ to 1, 2, 3, 5, 7, and 10, respectively, to study its impact on the performance of ADMPO-OFF. In addition, since ADM is an improvement for a fixed multi-step dynamics model (Asadi et al., 2018; 2019; Che et al., 2018), we further set the learning and roll-out steps of ADM to fixed lengths to highlight the importance of the any-step design. Table 10 reports the ablation results on four D4RL (Fu et al., 2020) MuJoCo datasets (v2 version). The network structure is kept unchanged.

Table 10: Normalized scores corresponding to different maximum backtracking lengths.

| Task Name | fixed 2-step | fixed 3-step | fixed 4-step | fixed 5-step | $m=1$ | $m=2$ | $m=3$ | $m=5$ | $m=7$ | $m=10$ |
|---|---|---|---|---|---|---|---|---|---|---|
| hopper-medium | 6.7±1.4 | 9.2±1.0 | 8.6±1.6 | 8.8±1.3 | 5.3±1.7 | 25.2±5.0 | 106.81±0.7 | **107.4±0.6** | 106.1±0.7 | 106.8±0.7 |
| hopper-medium-replay | 22.4±1.6 | 25.3±1.2 | 26.3±1.5 | 22.3±1.9 | 23.3±1.6 | 31.8±1.1 | 99.2±0.7 | **104.4±0.4** | 102.3±0.8 | 103.8±1.3 |
| walker2d-medium | 6.3±1.6 | 5.7±1.5 | 5.5±1.6 | 5.3±1.8 | 5.5±1.3 | **93.6±16** | 86.6±16.7 | 93.2±1.1 | 90.5±1.1 | 84.23±2.0 |
| walker2d-medium-replay | 17.4±1.6 | 24.5±2.2 | 21.9±2.6 | 22.6±2.1 | 19.8±2.7 | 66.3±0.7 | 83.7±0.6 | **95.6±2.1** | 94.3±0.8 | 91.4±1.0 |

When setting $m$ to 1 or using a fixed multi-step model, performance severely degrades. This is because, in these cases, state predictions lack diversity and cannot estimate model uncertainty, which is critical in the offline setting. Starting from 1 and gradually increasing $m$ to 5 leads to continuous performance improvement. Larger values of $m$ no longer bring further performance advantages since predicting across too many steps poses a challenge to the expressive capacity of neural networks. This experiment demonstrates the necessity of the any-step design and the proper selection of $m$.

### E.8 COMPUTING RESOURCE EVALUATION

In Table 11, we compare the computing resource consumption of ADMPO-OFF and MOPO, including the size of model parameters, GPU memory usage during training, and runtime.

The model parameter size of ADMPO-OFF is smaller than that of MOPO, primarily because MOPO uses an ensemble for its dynamics model. The GPU memory usage of ADMPO-OFF primarily depends on the value of $m$, and it is generally higher than that of MOPO. The runtime is roughly the same for both.

## F LIMITATIONS

In general, our work's limitations are summarized as follows.

- Similar to most previous MBRL algorithms focused on locomotion tasks, the dynamics modeling approach and uncertainty estimation used by ADMPO are likely not suitable for highly stochastic environments, as they are prone to being influenced by randomness, leading

Table 11: Comparison of ADMPO-OFF and MOPO in terms of computing resource consumption.

| Task Name | Size of Parameters (MB) | | GPU Memory (GB) | | Runtime (s/epoch) | |
|---|---|---|---|---|---|---|
| | ADMPO-OFF | MOPO | ADMPO-OFF | MOPO | ADMPO-OFF | MOPO |
| hopper-random | 3.77 | 7.66 | 7.85 | 3.46 | 10.20 | 7.55 |
| halfcheetah-random | 3.83 | 7.91 | 3.98 | 3.54 | 8.59 | 8.37 |
| walker2d-random | 3.83 | 7.91 | 3.98 | 3.54 | 9.78 | 8.19 |
| hopper-medium | 3.77 | 7.66 | 7.85 | 3.46 | 9.94 | 8.16 |
| halfcheetah-medium | 3.83 | 7.91 | 3.98 | 3.54 | 6.65 | 8.41 |
| walker2d-medium | 3.83 | 7.91 | 9.06 | 3.54 | 10.61 | 8.54 |
| hopper-medium-replay | 3.77 | 7.66 | 7.85 | 3.46 | 9.51 | 8.16 |
| halfcheetah-medium-replay | 3.83 | 7.91 | 3.98 | 3.54 | 7.35 | 8.27 |
| walker2d-medium-replay | 3.83 | 7.91 | 9.06 | 3.54 | 7.01 | 8.15 |
| hopper-medium-expert | 3.77 | 7.66 | 3.38 | 3.46 | 8.61 | 8.01 |
| halfcheetah-medium-expert | 3.83 | 7.91 | 3.98 | 3.54 | 8.49 | 8.07 |
| walker2d-medium-expert | 3.83 | 7.91 | 4.32 | 3.54 | 7.01 | 8.09 |

to poor performance. Improving ADMPO to address the challenges of highly stochastic environments is an important issue for future research. We will consider this as part of our future work.

- The computing resources used by ADMPO are mainly influenced by the maximum backtracking length $m$. When $m$ takes particularly large values, ADMPO requires significantly more GPU memory compared to the previous MOPO algorithm.

