# OpenReview forum: "Any-step Dynamics Model Improves Future Predictions for Online and Offline Reinforcement Learning"
_ICLR.cc/2025/Conference — ICLR 2025 Poster_

### Official Review · Reviewer_i58Z · 2024-10-31

**Soundness:** 2
**Presentation:** 2
**Contribution:** 2
**Rating:** 5
**Confidence:** 4

**Summary:**

Most convention rollout approaches in MBRL make prediction of the future state based on the current state alone. The main drawback is on the potential error accumulation as the prediction horizon grow. This work proposes Any-step Dynamics Model (ADM)  to mitigate this issue by making prediction based on previous state and actions information. This is achieved by using RNN structure to capture the dependence relationship among steps. Based on ADM, both online and offline compatible algorithms, i.e., ADMPO-ON and ADMPO-OFF, are introduced. Extensive experiments on MuJoCo and D4RL tasks show that the proposed method achieve better performance comparing with baseline methods. Meanwhile, it shows that ADM also help with the uncertainty estimation, where previous works normally use the ensemble method. The ablation studies on the rollout length $m$ is also validated in the experiment.

**Strengths:**

- Novel method on addressing bootstrapping errors during rollout. The proposed ADM incorporate an RNN structure to capture the dependencies among previous actions and state information, which can be helpful to avoid the prediction error induced by making predictions based on prediction. The ADM model is versatile and can be used for many offline and online policy optimization MBRL algorithms.
- The experiments on both online and offline environments. This work considers both online and offline experiments on verifying the performance. Meanwhile the studies on the uncertainty estimation also demonstrate that the proposed ADM can be a good alternative of ensemble method for estimation.

**Weaknesses:**

- The motivation of the proposed ADM rollout should be clarified further. In the proposed rollout in ADM (Algorithm 1), when predicting the future state such as $s_{m+1}$, $k=2$, it essentially is based on the information of $s_{m-1},a_{m-1:m}$. When $s_{m}$ is available in the given sequence, it seems not clear why the proposed ADM can perform good. Moreover, when the state transition and policies are stochastic, the proposed rollout essentially requires to consider the stochasticity in $s_m$ and $a_m$ when making prediction for $s_{m-1}. What is the intuition of using this rollout for the sake of the performance.
- The proposed rollout algorithms also make prediction based on the previous prediction, e.g., when predict $s_{m+2}$, it may require $s_{m+1}$ (which is obtained in the previous steps).  It is thus very important to clarify why this method can effectively mitigate the bootstrapping errors comparing with the conventional approach.
- The backtracking length $m$ is a newly introduced parameter, and the ablation studies show that it affects the performance a lot. How to choose such a parameter in practice for specific task? Will it work for all different tasks?

**Questions:**

- This work mentions about MBPO (Janner et al. 2019), where they also proposed a truncated rollout to address the error accumulation over long horizon. Many works on adaptive rollout length should be addressed and compared [Q1,Q2].
- line 046, what is deviation error? Line 68, what do you mean by trajectory distribution?
- What is the computation complexity of the proposed method?



[Q1] Frauenknecht, Bernd, et al. "Trust the Model Where It Trusts Itself-Model-Based Actor-Critic with Uncertainty-Aware Rollout Adaption." ICML.

[Q2] Zhang, Weinan, et al. "Model-based multi-agent policy optimization with adaptive opponent-wise rollouts." arXiv preprint arXiv:2105.03363 (2021).

**Details Of Ethics Concerns:**

The proposed method is a general RL method.

---

### Official Review · Reviewer_NWuZ · 2024-10-31

**Soundness:** 2
**Presentation:** 2
**Contribution:** 2
**Rating:** 5
**Confidence:** 3

**Summary:**

The authors propose a dynamics model that is trained to predict a future state and reward tuple given various length state-action sequences. Notably the input to the model is not is not $(s_{t-3}, a_{t-3}, s_{t-2},a_{t-2},s_{t-1},a_{t-1})$ as one would expect, but instead they are using the same state, i.e. $(s_{t-3}, a_{t-3}, s_{t-3},a_{t-2},s_{t-3},a_{t-1})$.
This somehow leads to a signficiantly better performance and improved results when used for model-based online and Offline RL.
For Offline RL, the authors also use the ensemble disagreement for different input-lenghts as a measure of model uncertainty.

**Strengths:**

* The results look quite promising, performing well in Online RL and Offline RL benchmarks

**Weaknesses:**

* Clarity: The paper is overall hard to follow with poor writing and inconsistent notation. For example, sometimes $\hat{s}$ is used to refer to states predicted by the dynamics model, sometimes $s$ is used to refer to predicted states. $s$ is also used to refer to ground-truth states. In addition to various typos this makes the paper hard to follow.
 * Soundness: The paper does not give a convincing reason for why their proposed method works better, nor does it include ablations or experiments that could elucidate why it performs well. There also isn't any theoretical analysis that might be helpful to understand it better.

Some important implementation choices are not justified either. To list the most major issues:
 * As written in L152, and shown in Figure 1, the state $s_t$ is repeated as input for all GRU/ model steps. This is reasoned to prevent error accumulation of using model predictions as input. However the paper assumes access $m$ step sequences which are then used to predict the next $H$ steps. Why aren't all $m$ steps used as model inputs?
 * Using a random input length during training is reasonable to learn a model that performs well for different input lenghts. However, when using the model to train a policy it would seem more natural to average across lenghts to obtain a better prediction, or choose the longest length. Why is a random length used?

* Experiments: Hyper-parameter choice is not discussed and only 5 seeds are used. For most baselines, the results were copied from the original papers. As hyper-parameter choice and implementation details are important for RL methods, this is not sufficient, especially when applying the algorithms to a new benchmark (NeoRL).


Minor issues:
 * Theorem 3.5 follows rather straightforwardly from Assumption 3.3 and does not provide any new insights. It is also not specific to the ADM and could apply to any model-based RL method.
 * parantheses should be placed around the author name for citations, not just around the year number.
 * What the authors refer to as "bootstrapping prediction" issues seems more like error accumulation to me.  Using the term bootrstrap can be a bit confusing in an RL context, but this is a minor point.
* Algorithm 2 and 1 should probably combined and both be in the main text, as they are important to understand the approach.
* L241: "explore beyond the boundaries of the risky regions" should be "beyond the boundaries of the safe regions"?
* L352: should this be $\hat{s}_{t+m}$? If not, there can be no compounding error.
 *

**Questions:**

* Why aren't all $m$ known states used as input for the model?
 * Why is a random $k$ used to train the policy?
 * Fundamentally, this paper is missing a good explanation for why its method works better. Additional ablations would help elucidate this. For example, inputting the whole known state sequence in to the GRU instead of repeating the first state. Or only inputting the state into the first GRU-call should have the same information content.

Overall I believe this paper has potential to be very useful, but it requires additional work to improve the presentation and explanation of its method before publication.


After rebuttal:

The authors have provided helpful experiments that confirm their changes to the model to be advantageous.
I still have concerns particularly about the lack of hyper-parameter optimization for baselines.

Overall I am raising my score (3->5)

---

### Official Review · Reviewer_eBkb · 2024-11-01

**Soundness:** 2
**Presentation:** 3
**Contribution:** 2
**Rating:** 6
**Confidence:** 3

**Summary:**

The authors propose Anytime Dynamics Models, which are trained to predict k-steps-ahead states and rewards given a past state and a sequence of actions. ADM are designed to reduce bootstrapping in autoregressive predictions, and alleviate accumulating errors over long horizons. ADM can be directly utilized for model-based policy optimization, and lead to improved sample efficiency in standard mujoco tasks. This work also proposes an estimate for the mixture of aleatoric and epistemic uncertainty by computing the disagreement of the model's predictions for different number of steps. Uncertainty estimates can then replace those obtained through model ensembling in the MOPO framework, again showing improvements in performance on D4RL and NeoRL tasks.

**Strengths:**

- The presentation is clear and the paper is well written.
- The empirical evaluation is broad, and the proposed methods largely outperform the selected baselines.

**Weaknesses:**

- The main motivation of this work is that of alleviating autoregressive error accumulation. Nevertheless, ADMs rely on a recurrent architecture that implicitly bootstraps on previous hidden states. It is not entirely clear to me why this would not also result in error accumulation. Empirical performance provides some support to the method's effectiveness, but are there any principled reasons why ADMs suffer less from error accumulation? In my opinion, this one of the main issues with the current version of this work. The authors report a Theorem (3.4), but it appears to be a general result. How does it relate to the backtracking nature of the proposed method?
- The second issue is in the limited comparison to recurrent forward models. Recurrent world models are at the core of SOTA model-based algorithms [1,2]. The only empirical comparison to recurrent models is, as far as I understand, in Figure 2, which is not easy to interpret. The RL evaluation does not include any recurrent architecture. This raises the following question: is the improvement in RL performance due to any-time dynamic sampling, or to the choice of architecture? For instance, how would the method compare in online RL to a RSSM [1] of similar size, using standard autoregressive sampling?
- A third main issue is an insufficient discussion of the limitations of this method. A fundamental issue which is not mentioned in Section 6 is that the uncertainty estimation method proposed appears to mix aleatoric and epistemic uncertainty. This is in general suboptimal: how would if perform in an environment in which the optimal policy traverses stochastic parts of the MDP?

**Questions:**

Three questions are raised as part of the weaknesses. I would additionally ask the following:
- Figure 2 shows that error do indeed accumulate less drastically in ADMs. However, the scale of the plots is not meaningful. An error of 10^34 is probably not good for planning, but the error incurred by ADMs might as well be too high. This is hard to see due to the y-scale of the plots. Furthermore, it's not clear what the scale of the states is, and a relative measure of error might be more informative. Can the authors provide a more readable plot? Moreover, Which policy is used during the rollouts?
- The idea of temporally consistency in any-step predictions as a proxy for epistemic uncertainty has been explored in the past. Among other works, [3] uses uncertainty of the value, which prevents from mixing aleatoric and epistemic uncertainty. These previous works should be acknowledged.
- ADMPO tunes hyperparameters on each task. How are the hyperparameters for the baselines selected?
- There are inconsistencies in the spelling of the word rollout/roll-out.
- Equation 1: for the sake of clarity, can the authors specify over which spaces/sets the sum is computed?
- line 204: what about these algorithms is "foundational"?

**References:**

[1] Hafner et al., Mastering Diverse Domains through World Models, arXiv 2023

[2] Hanses et al., TD-MPC2: Scalable, Robust World Models for Continuous Control, ICLR 2024

[3] Filos et al., Model-Value Inconsistency as a Signal for Epistemic Uncertainty, ICML 2022

---

> ### Comment · Reviewer_eBkb · 2024-11-21
> **Thank you for the response!**
>
> I would like to thank the authors for answering my comments. I understand that the benefits of multi-step dynamic modeling stem from avoiding autoregressive predictions. In a way, if I understand correctly, the proposed any-step dynamic model uses an RNN to predict a $k$-step-ahead state. This RNN receives a sequence of $k$ actions, and thus updates its hidden state $k$ times (Figure 1). Is this correct? If that is the case, could the author explain why error accumulation does not occur in this setting, but does occur when explicitly predicting intermediate states?
>
> As my remaining concerns on hyperparameter tuning and limitations were addressed, I am happy to raise my score to a 6.

---

> ### Comment · Reviewer_eBkb · 2024-11-23
>
> Thank you for pointing out the oversight and for providing the remaining clarifications. The score should now also be updated.

---

### Official Review · Reviewer_pnqF · 2024-11-10

**Soundness:** 3
**Presentation:** 3
**Contribution:** 3
**Rating:** 8
**Confidence:** 4

**Summary:**

This paper introduces the Any-Step Dynamics Model (ADM), an RNN-based dynamics model with a variable prediction horizon. ADM enhances the fidelity of dynamics modeling and provides uncertainty estimates by leveraging variability across different prediction horizons.

**Strengths:**

[S1] The approach to uncertainty estimation is novel, contrasting traditional ensemble methods by using variance across horizons within a single model instead of variance across the prediction of multiple models.

[S2] ADM demonstrates strong performance in both online and offline experiments, with uncertainty measures that correlate well with ground-truth errors.

**Weaknesses:**

[W1] ADM uses multiple-step rollouts, initializing from a state several steps back in time to predict the trajectory, thus potentially disregarding states closer to the current time step. This may increase compounding errors in rollouts compared to ensembles, which operate with single-step rollouts based on the current state.

[W2] The reference formatting in the main text appears incorrect; minor revisions are recommended.

**Questions:**

[Q1] ADM currently utilizes 2 to 5 backtracking steps. What would happen with a larger number? Would increased steps result in model divergence due to compounding errors? (See Weakness 1.)

[Q2] Could the multi-step rollout compounding error be mitigated? For example, including the ground-truth state $ s_{t+k} $ instead of only $s_t$ in this work for each roll-out step in an autoregressive manner, as done in models like the Trajectory Transformer, may improve accuracy. (See Weakness 1.)

---

### Meta-Review · Area_Chair_tjEf · 2024-12-19

**Metareview:**

This paper proposes the Any-step Dynamics Model (ADM), which is a model-based method that learns to predict transitions multiple steps into the future to avoid accumulating errors due to bootstrapping. They provide an online and an offline variant of their method, with their efficacy demonstrated by empirical evaluations. Additionally, the authors provide a mechanism for uncertainty estimation using ADM in the offline setting.

The paper is clear and well-written, and the empirical evidence supports the claims.

The main criticism from reviewers seems to revolve around hyper-parameter optimization for the baselines, in particular in the NeoRL benchmark, as the baselines were likely not optimized for that benchmark. Given that most of the evaluations were done in more standard (and well-understood) benchmarks, I believe it is reasonable to use the hyper-parameters provided by the original algorithm implementations. It would be good if the authors could make this point more clearly for readers.

Although there are some concerns on how it can handle stochastic environments, many works do introduce novel ideas in systems with limited stochasticity, so I do not see this as a reason to reject.

Thus, I am recommending acceptance of this work, as it does introduce a novel algorithm with reasonable empirical evidence for its effectiveness, which may be of general interest to the RL community.

**Additional Comments On Reviewer Discussion:**

There were a number of concerns raised on issues with error accumulation, but the authors clarified most of these and improved their submissions.
The issues with regards to hyper-parameter selection and stochasticity remained mostly unaddressed (excluding the comments added to the limitations section).

See my comments above for my thoughts on these.

---

### Decision · Program_Chairs · 2025-01-22

Accept (Poster)